# Transcriptomic sex differences in early human fetal brain development
Federica Buonocore [1] ✉, Jenifer P. Suntharalingham [1], Olumide K. Ogunbiyi[2,3], Aragorn Jones [4], Nadjeda Moreno[3], Paola Niola[5], Tony Brooks[5], Nita Solanky[3], Mehul T. Dattani [1], Ignacio del Valle [1,6] & John C. Achermann [1,6]

The influence of sex chromosomes and sex hormones on early human brain development is poorly understood. We therefore undertook transcriptomic analysis of 46,XY and 46,XX human brain cortex samples ($n = 64$) at four different time points between 7.5 and 17 weeks post conception (wpc), in two independent studies. This developmental period encompasses the onset of testicular testosterone secretion in the 46,XY fetus (8wpc). We show differences in sex chromosome gene expression including X-inactivation genes (*XIST*, *TSIX*) in 46,XX samples; core Y chromosome genes ($n = 18$) in 46,XY samples; and two Y chromosome brain specific genes, *PCDH11Y* and *RP11-424G14.1*. *PCDH11Y* (protocadherin11 Y-linked) regulates excitatory neurons; this gene is unique to humans and is implicated in language development. *RP11-424G14.1* is a long non-coding RNA. Fewer differences in sex hormone pathway-related genes are seen. The androgen receptor (*AR*, NR3C4) shows cortex expression in both sexes, which decreases with age. Global cortical sex hormone effects are not seen, but more localized AR mechanisms may be important with time (e.g., hypothalamus). Taken together, our data suggest that limited but potentially important sex differences occur during early human fetal brain development.

The human brain undergoes remarkable growth and differentiation during the first and second trimester[1,2], but many of the mechanisms that influence these processes remain poorly understood. In recent years, rapid progress has been made in the development of techniques that can analyze gene and protein expression in the brain across the lifespan, and an increasing number of resources are becoming available to share data and promote scientific progress[3–6]. One major area of interest has been the potential impact of biological sex differences on neurobiological development and function[7]. However, in humans, most reported studies in this area have focused on post-natal or adult life, and surprisingly few data are available related to sex differences during critical early stages of brain development[8].

Early developmental sex differences can potentially result from the influences of sex chromosomes as well as the effects of sexually dimorphic sex hormones in early life. A complex interplay between these elements and other factors may exist[7,9].

The sex chromosome complement is established at the time of fertilization as a result of sex chromosome combination (46,XX typically in girls; 46,XY typically in boys)[10]. The X chromosome encodes >800 genes and has a small number of genes with Y chromosome homologues in the pseudoautosomal regions. In contrast, the Y chromosome encodes for only around 60-100 genes that play a key role in sex development and fertility, as well as more diverse biological functions[11,12].

Sex hormones are generally synthesized and released by the developing gonads (testes, ovaries). In the 46,XY embryo, the Y chromosome gene *SRY* (OMIM 480000) is expressed in the bipotential gonad at around 6 weeks post conception (wpc), which triggers a cascade of downstream genes leading to testis determination[13–15]. We have previously shown by modeling time-series analysis of gene expression that maximal upregulation of the enzymes needed to synthesize testosterone occurs at 8wpc in the human testis[13]. Fetal testicular testosterone acts to stabilize Wolffian structures (e.g., seminal vesicles, vas deferens) and is converted to the more potent hormone,

[1]Genetics and Genomic Medicine Research and Teaching Department, UCL Great Ormond Street Institute of Child Health, University College London, London, WC1N 1EH, UK. [2]Department of Histopathology, Great Ormond Street Hospital for Children National Health Service (NHS) Foundation Trust, London, WC1N 3JH, UK. [3]Developmental Biology and Cancer Research and Teaching Department, UCL Great Ormond Street Institute of Child Health, University College London, London, WC1N 1EH, UK. [4]Biosciences Institute, Faculty of Medical Sciences, Newcastle University, Newcastle upon Tyne, NE2 4HH, UK. [5]UCL Genomics, Zayed Centre for Research, UCL Great Ormond Street Institute of Child Health, University College London, London, WC1N 1DZ, UK. [6]These authors contributed equally: Ignacio del Valle, John C. Achermann. ✉e-mail: f.buonocore@ucl.ac.uk

dihydrotestosterone (DHT) by the enzyme 5α-reductase type 2 (*SRD5A2*; OMIM 607306) in the external genital region to promote penile and scrotal growth[16]. Testosterone acts exclusively through the androgen receptor (*AR*) (also known as *NR3C4*; OMIM 313700) and can be converted to estrogens by the enzyme aromatase (*CYP19A1*; OMIM 107910). Most data suggest that the developing ovary in the 46,XX fetus is endocrinologically quiescent at this time and does not secrete significant amounts of the female-typical hormone, estradiol. Thus, the late first trimester (from 8wpc) and early second trimester is a critical stage in human maturation and endocrine effects.

Sex chromosomes and sex hormones play important dynamic roles during development, leading to longer term effects after birth, at puberty and in adult life, for example, shaping the way that specific conditions may affect males and females differently. These affect not just reproductive and endocrine conditions, but also influence many biological processes such as cardiovascular, immune, and neurological function. For example, in the field of clinical neuroscience, autistic spectrum disorder (ASD), conduct disorders, schizophrenia and Parkinson's disease are more common in males, whereas anxiety disorders and dementia are typically more prevalent in females[17]. Sex differences can therefore potentially play a central role both in disease mechanisms and in the way specific conditions manifest.

Several previous studies looking at sex differences in development have focussed on animal models[7,9,18–21]. Studies of human fetal brain development are relatively limited and often include only a small fetal or early postnatal group as part of a life course study spanning many ages. For example, Weickert et al. used microarray analysis and quantitative PCR to examine prefrontal cortex of postnatal human brain samples (1 month to 50 years) and reported several developmental changes in transcripts between males and females[22]. Reinius & Jazin and Kang HJ, et al. both studied different regions of the prenatal human brain by microarray and found that largest sex differences were due to Y chromosomes genes[23,24]. Shi et al. analysed major developmental stages (prenatal to adulthood) from the BRAIN SPAN atlas (RNAseq data) and reported consistent expression of twelve Y chromosome genes across all stages[25].

In order to specifically study the influence of sex chromosomes and sex hormones, and further address sex differences in early human brain cortex development, we undertook transcriptomic analysis across a critical time period of development between 7.5-8wpc and 15–17wpc, using two independent datasets of bulk RNA-sequencing (*n* = 32 each), at a time from just before the onset of fetal testicular testosterone secretion and in subsequent weeks. The main aims of the study were to address whether: (1) global differences exist between 46,XX and 46,XY samples that could represent the effect of sex chromosome-related genes, and whether this differs in the brain compared to other tissues; (2) whether any differentially expressed genes have strong brain specificity; (3) whether sex-related divergent patterns in global cortical gene transcription occur over time and whether these could potentially represent sex-hormone (testosterone)-dependent events; and (4) whether more localized sex hormone effects might occur in the developing brain cortex or other regions that could be linked to biological outcomes.

## Results

### Global sex differences in gene transcription during early human brain cortex development

In order to investigate potential global sex differences in gene transcription in brain development, 32 samples were obtained from developing human brain telencephalon/cortex between Carnegie Stage 22-23 (CS22-CS23, corresponding to 7.5-8wpc) and 15-17wpc (termed "Brain-Seq 1") (Fig. 1a). An overview of the study design is shown in Table 1, which included matched 46,XX and 46,XY samples at each of four different stages (CS22-23, 9wpc, 11-12wpc, 15–17wpc). This time period represents a key phase in brain cortex development and differentiation, and follows the onset of testosterone secretion by the 46,XY testis from around 8wpc (Fig. 1b, c).

A bulk RNA-Seq dataset with matched samples was also generated using the same bioinformatic pipeline from raw data available as part of a Human Developmental Biology Resource (HDBR) fetal brain transcription repository (Table 1; see Methods)[26]. This dataset (termed "Brain-Seq 2") allowed a parallel, independent replication and validating study across the same developmental time course, using standardized protocols.

Principal component analyses (PCA) for both Brain-Seq 1 and Brain-Seq 2 datasets are shown in Fig. 1d. Principle component (PC) 1 generally reflected developmental stage of the sample (Brain-Seq 1, 35% variance; Brain-Seq 2, 42% variance), whereas the second component (PC2) likely reflected karyotype of the sample and any potential sex differences in transcription (Brain-Seq 1, 19% variance; Brain-Seq 2, 20% variance). However, analysis of bulk-RNA-seq from other development tissues (e.g. kidney, pancreas, liver, skin) also showed sample distribution related to karyotype, suggesting that sex differences due to sex chromosomes are not unique to the brain cortex but are common also to other tissues during early development (Supplementary Fig. 1).

### Global differential gene expression

To investigate differences in gene expression between 46,XX and 46,XY samples, we carried out differential expression analysis between all 46,XX samples and all 46,XY samples in each RNA-seq dataset (Fig. 2; Supplementary data 1.14_Brain-Seq 1 and 2.14_Brain-Seq 2). Using a cut-off of adjusted *p*-value of < 0.05, four genes involved in X chromosome inactivation showed consistently higher expression in all 46,XX samples in both datasets (*XIST, TSIX, JPX, ZFX*) (Fig. 2a), and twenty genes showed consistently higher expression in all 46,XY samples in both datasets (Fig. 2b). We observed a significant correlation between all genes in Brain-Seq 1 and Brain-Seq 2 (Fig. 2c), and a stronger correlation when only those differentially expressed genes (DEGs) with an adjusted *p*-value of < 0.05 were considered (Fig. 2d).

### Stage-specific analyses of gene expression

To analyze gene expression patterns in more detail across developmental stages, differential expression of key sex chromosome-related genes was visualized in volcano plots of 46,XX versus 46,XY samples at each stage and in both Brain-Seq 1 and Brain-Seq 2 datasets (Fig. 3; Supplementary data 1.2-1.9_Brain-Seq 1; Supplementary data 2.2-2.9_Brain-Seq 2). Using a cut-off of adjusted *p*-value of < 0.05, only two genes showed consistently higher expression in 46,XX samples at all age stage comparisons; these were the two key regulators of X inactivation, *XIST* and *TSIX* (Fig. 4a, c, e). In contrast, a "core" group of 18 genes were consistently highly expressed in 46,XY brain samples, which were all Y chromosome genes (Fig. 4b, d, f).

### Brain-specific sex differences in gene transcription

In order to identify brain-specific genes during development, differentially expressed 46,XY genes (46,XY versus 46,XX, adjusted *p*-value < 0.05) in the brain cortex (Fig. 4f) were compared with differentially expressed 46,XY genes in the other tissues (kidney, pancreas, liver, skin) (Fig. 5a; Supplementary data1_Brain-Seq 1; Supplementary data2_Brain-Seq 2; Supplementary data3_Controls). Using this approach only two genes emerged as showing strong brain specific expression (*PCDH11Y, RP11-424G14.1*). *PCDH11Y* (OMIM 400022) encodes protocadherin 11 Y linked, a cadherin-family extracellular adhesion molecule involved in cell-cell communication in excitatory neurons (https://www.proteinatlas.org/ENSG00000099715-PCDH11Y). Significantly higher expression was confirmed in the developing 46,XY brain compared to 46,XX brain and to 46,XY control tissues by more detailed analysis of the bulk RNA-seq data and by qRT-PCR (Fig. 5b–d). These time series-analyses of the bulk RNA-seq datasets and qRT-PCR revealed a marked increase in *PCDH11Y* in the 46,XY brain with age, especially at 15–17wpc (Fig. 5b, d). Consistent with this is the differential expression of *PCDH11Y* in the adult brain, compared to other tissues, in Human Protein Atlas Consensus and Genotype-Tissue Expression (GTEx) data (Fig. 5e, Supplementary Fig. 2).

*PCDH11Y* is located on the short arm of the Y chromosome (p11.2) (Chr Y: 4,868,267-5,610,265, GRCh37; Chr Y:5,000,226-5,742,224,

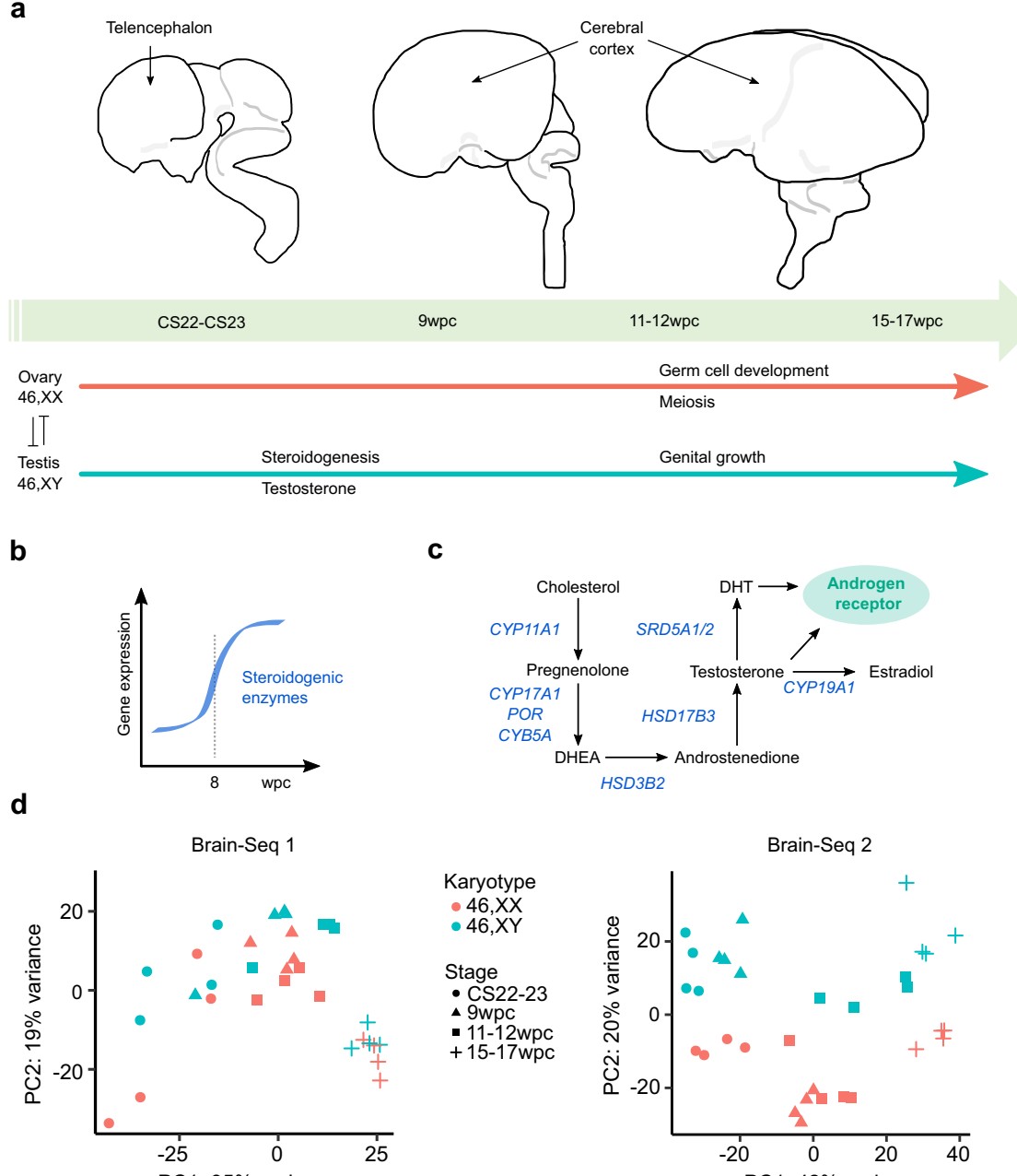

**Fig. 1 | Global sex differences during early brain development. a** Model of human brain development in relation to gonad development. Telencephalon and cerebral cortex regions of the developing brain are indicated by arrows. Gonad determination into either testis or ovary begins at around Carnegie Stage 18 (CS18) (6 weeks post conception, wpc). Testicular testosterone synthesis and secretion occurs from around CS23 (8wpc) in the 46,XY fetus. **b** Gene expression of factors implicated in testicular testosterone biosynthesis is upregulated at 8wpc. Data derived from del Valle et al.[13]. **c** Simplified pathway of androgen biosynthesis with genes encoding enzymes showed in blue. DHEA, dehydroepiandrosterone; DHT, dihydrotestosterone. **d** Principal component analysis (PCA) for each independent bulk RNA-seq dataset (Brain-Seq 1, Brain-Seq 2) included in the study showing all samples (total *n* = 32 in each dataset, see Table 1) based on the first two principal components (PC1, PC2).

GRCh38) in a locus with strong X chromosome homology ("X-degenerate region"), that is separate from the pseudoautosomal regions. Although *PCDH11Y* is unique to humans, a related X chromosome gene (*PCDH11X*) exists (Supplementary Fig. 3). In both our Brain-Seq 1 and Brain-Seq 2 datasets, *PCDH11X* expression was similar in both 46,XX and 46,XY samples (Fig. 5f). *PCDH11X* was seen in the brain and some other tissues in GTEx, with no clear sex differences (Supplementary Fig. 4). Thus, the Y chromosome gene *PCDH11Y* likely has an additive or unique effect in the developing and post-natal 46,XY brain.

The other brain-specific differentially expressed transcript identified using this analysis pipeline was a long non-coding RNA, *RP11-42G14.1* (also referred to as ENSG00000260197) (Yq11.222) (Chr 12: 76,357,812-76,358,780, GRCh37; Chr Y:19,691,941-19,694,606, GRCh38.p14) (Supplementary Fig. 5). The gene with closest proximity to *RP11-424G14.1* is *KDM5D* (Chr Y: 21,865,751-21,906,825, GRCh37; Chr Y:19,703,865-19,744,939, GRCh38), but both have a minus strand orientation and *RP11-424G14.1* is 3' to *KDM5D* (Supplementary Fig. 5). Little is known about the putative function of this gene.

**Table 1 | Overview of all samples used in the study**

| Developmental stage | Brain-Seq 1 dataset | | | Brain-Seq 2 dataset | | |
|---|---|---|---|---|---|---|
| | 46,XX (n) | 46,XY (n) | Total (n) | 46,XX (n) | 46,XY (n) | Total (n) |
| CS22 | 4 | 4 | 8 | – | 3 | 3 |
| CS23 | – | – | – | 4 | 1 | 5 |
| 9wpc | 4 | 4 | 8 | 4 | 4 | 8 |
| 11wpc | 4 | 4 | 8 | 4 | – | 4 |
| 12wpc | – | – | – | – | 4 | 4 |
| 15wpc | 3 | 3 | 6 | – | – | – |
| 16wpc | 1 | 1 | 2 | 4 | – | 4 |
| 17wpc | – | – | – | – | 4 | 4 |
| Overall | | | 32 | | | 32 |

Description of brain cortex samples analyzed in our study, showing the number of 46,XX and 46,XY samples per each developmental stage. CS22 corresponds to 7wpc + 4 d (days); CS23 corresponds to 8wpc. *n* number, *wpc* weeks post-conception.

## Brain-enriched Y chromosome genes

In addition to "brain-specific" genes identified using this approach, we also investigated the potential differential expression of "brain-enriched" genes from the Y chromosome. These genes demonstrated differential expression in the developing 46,XY brain but also more variable degrees of differential expression in different control tissues (46,XY versus 46,XX), meaning they did not reach the threshold for detection in our pipeline based on adjusted *p*-value. Nevertheless, these genes could still have an important role in development.

The main genes identified were *ANOS2P, DDX3Y, EIF1AY, GYG2P1, KDM5D, NLGN4Y, PRKY, RSP4Y1, TBL1Y, TMSB4Y, TTTY14, TXLNGY, USP9Y, UTY, and ZFY*, with remarkable consistency between both Brain-Seq 1 and Brain-Seq 2 datasets. As expected, these genes fall within the "core" set of 18 genes and are represented within the context of other Y chromosome genes in Supplementary Fig. 6. We therefore explored expression of these genes across tissues in more detail using the Human Protein Atlas Consensus RNA expression panel of 50 different tissues (including 10 neuronal-related regions), but no genes with high brain specificity were identified, although *NLGN4Y* shows brain expression[27,28] as well as expression in multiple other tissues (Supplementary Table 1). *PCDH11Y* emerged as the Y gene with highest brain specificity.

## Sex hormone effects on global early human brain development

In addition to differences in sex chromosome-related transcripts, differences in sex hormones and their pathways are also likely to influence sex dimorphic aspects of brain development during gestation.

As outlined above, the 46,XY human embryo/fetus develops testes that start to synthesize and release androgens (testosterone) into the developing blood stream from around 8wpc (Fig. 1b)[13]. Testosterone typically acts through the androgen receptor (*AR, NR3C4*), and in some tissues such as the developing genital tubercle/external genitalia, testosterone has to be converted to the more potent androgen, dihydrotestosterone (DHT) (Fig. 1c). The role of estrogens (e.g., estradiol) in the developing brain is unclear, although the ovary is not thought to synthesize significant amounts of estrogen during early development (Fig. 1c).

To address potential sex hormone pathways in the developing brain, we first analyzed *AR* gene expression across time in the Brain-Seq 1 and Brain-Seq 2 fetal brain cortex datasets. Both datasets showed a remarkably similar level of *AR* expression in 46,XY and 46,XX tissues, and a consistent and marked decrease in *AR* expression with age (Fig. 6a, b). This finding was consistent with generally low levels of *AR* expression in the adult brain, and globally similar *AR* expression in males (46,XY) and females (46,XX) in GTEx (Supplementary Fig. 7). In order to discover potential androgen responsive genes during this period, we looked for genes that showed increased differential expression (adjusted *p*-value < 0.05) in 46,XY samples

following testosterone exposure (9wpc vs CS22/23; 15-17wpc vs CS22/23), that were not increased in 46,XX time series data (Supplementary Figs. 8 and 9; Supplementary Data 1.10-1.13_Brain-Seq 1; Supplementary data 2.10-2.13_Brain-Seq 2). Ten genes were found that met these criteria in both the Brain-Seq 1 and Brain-Seq 2 datasets. However, time series expression analyses revealed that this differential expression was not consistent in just the 46,XY samples (Supplementary Fig. 10). Similarly, no clear potential androgen-responsive genes emerged when we added a function requiring significantly higher differential expression in 46,XY versus 46,XX tissue (Supplementary Fig. 11). We also sought potential AR co-factors by comparing 46,XX and 46,XY DEGs of both Brain-Seq 1 and Brain-Seq 2 datasets with a database of AR interacting proteins (Reactome KnowledgeBase)[29]; although several factors were differentially expressed in the developing brain (Supplementary Fig. 12), no consistent enrichment of AR interactors in the dataset was seen.

As it is unclear whether any potential androgen-driven effects in the fetal brain could be through a direct action of testosterone, or through the generation of DHT, we investigated the brain cortex expression of enzymes involved in "classic" and "non-classic/backdoor" pathways for DHT generation[30–32] (Supplementary Figs. 13 and 14). Very low expression of *SRD5A2* was seen, the enzyme in the classic pathway that generally coverts testosterone to DHT in reproductive tissues (e.g. developing penis, prostate) (Figs. 1c, 6c, d). *SRD5A1* and *SRD5A3* enzyme isoforms were expressed with similar levels in 46,XY and 46,XX cortex (Fig. 6c, d). The expression of other proteins and enzymes involved in the "classic" and "backdoor" pathways of androgen biosynthesis were extremely variable in the brain cortex, although an almost complete lack of *CYP17A1* expression (encoding 17α-hydroxylase/17,20-lyase) would likely be rate limiting; for example, in the potential neurosteroidogenesis of androgens from precursors such as placental progesterone or fetal adrenal gland androsterone[32–34] (Supplementary Figs. 13 and 14). Similarly, as estrogens are proposed to play a key role in brain sex differences in some species, we investigated these pathways further in our human fetal datasets. Expression of the gene encoding aromatase (*CYP19A1*) (needed to convert androgens to estrogens, Fig. 1c) and of the estrogen receptors (*ESR1, ESR2*) all had extremely low counts in all cortex samples and did not show any sex dimorphic differences (Supplementary Fig. 15).

## Potential localized androgen receptor expression

Given the detectable *AR* expression seen at a transcript level in both 46,XX and 46,XY brain cortex, we undertook immunohistochemistry using an AR specific antibody that was validated in human development. Although most areas of cortex did not show strong AR expression, clear regions of AR positivity were seen in the nuclei of cortical cells in early development (Fig. 7a–c). Furthermore, more localized regions of AR expression were seen including in midline structures and the basal hypothalamus (Fig. 7d, e and Supplementary Fig. 16).

In order to address this further, we analyzed several available resources that investigated transcriptomic expression during early human embryonic and fetal brain development (Table 2). Using the GTEx dataset in adult tissues across the lifespan, highest expression of *AR* was seen in the anterior pituitary and hypothalamus regions (Supplementary Fig. 7; see also https://www.proteinatlas.org/ENSG00000169083-AR/brain). We therefore focused on these regions in more detail, as they are also involved in hypothalamic-pituitary-endocrine regulation. Using a human fetal pituitary gland single cell RNA-sequencing (scRNA-seq) analysis resource, we identified highest *AR* expression in the corticotrope cell lineage, suggesting a potential role in the adrenal axis and later stress mechanisms[35] (Fig. 7f, Table 2, Supplementary Fig. 14a). Furthermore, in a human hypothalamus single cell dataset we were able to localize *AR* expression in several areas, most strongly in the arcuate nucleus (ARC)[36] (Fig. 7g, Table 2, Supplementary Fig. 16b). These findings suggest that localized expression of the *AR* may be important with time. In addition, we generated feature plots in both datasets (pituitary, hypothalamus) for *UTY* (to identify 46,XY cells) and *XIST* (to identify 46,XX cells). This approach

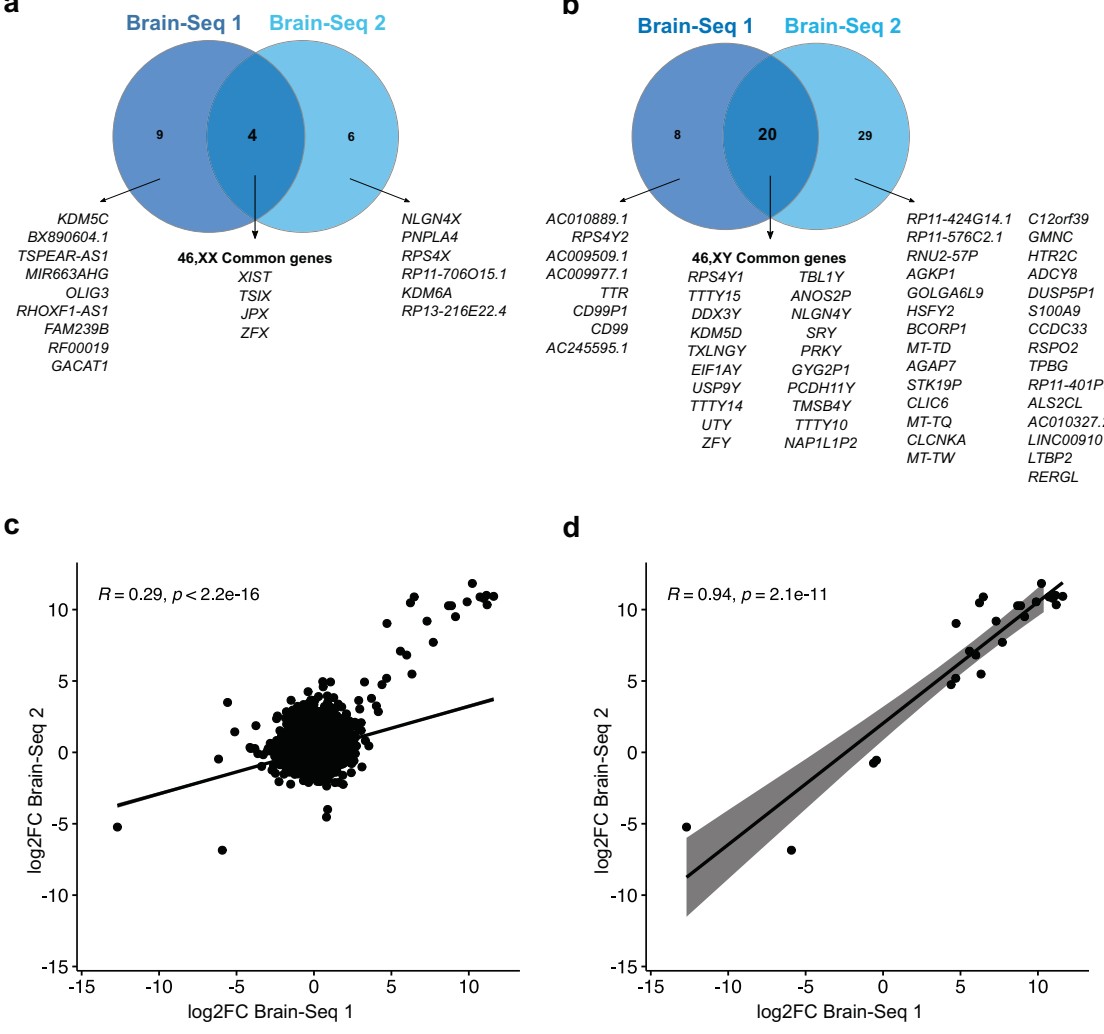

**Fig. 2 | Global differential gene expression patterns between 46,XX and 46,XY samples during early brain development.** Two independent RNA-seq datasets were analyzed (Brain-Seq 1 and Brain-Seq 2), each containing $n = 16$ 46,XX and $n = 16$ 46,XY samples across all developmental stages (CS22-23; 9wpc; 11-12wpc; 15–17wpc) (Table 1). **a, b** Venn diagrams showing the overlap of differentially expressed genes (DEGs) in both RNA-seq datasets at all developmental stages. **a** All 46,XX samples; (**b**) All 46,XY samples; (**c-d**) Scatterplots of DEGs in Brain-Seq 1 and Brain-Seq 2. **c** All DEGs; **d** DEGs with adjusted $p$-values < 0.05. Black line and gray shading, weighted Deming regression and 95% confidence interval. Pearson correlation coefficients and adjusted $p$-values are indicated.

confirmed the presence of both 46,XY and 46,XX cells, and a potential higher expression of *AR* in 46,XY-derived cells (Supplementary Figs. 17 and 18). Taken together, future studies should focus on more localized regions of high AR expression as well as on potential sex differences in specific cell populations.

## Discussion

Although sex differences in early development are potentially biologically important, relatively little is known about these at a genetic level in the late first trimester and early second trimester in humans. We therefore undertook this time-series study looking at potential effects of sex chromosomes and the machinery for sex hormone action across this time frame in the developing human cerebral cortex.

We intentionally chose to focus on a large, well-matched bulk RNA sequencing approach. Whilst this does not provide the granular detail of transcriptomics in single cells, it has the advantage of allowing detection of low-level gene expression and trends, especially in time series data, and has less sampling bias than might occur with a more focal single cell approach. In order to strengthen the robustness of this strategy, we included two matched and independent replication datasets of 32 samples each at similar time points (Brain-Seq 1 and Brain-Seq 2), which were balanced for 46,XY

and 46,XX samples. We also chose a study period spanning the developmental stages just before testicular testosterone secretion in the 46,XY fetus, and in the weeks after its onset, in order to assess any basal sex differences and any potential divergent changes in gene expression which could be due to differences in sex hormone action.

Our analysis initially focused on global differences in gene expression across this critical time period. Using principal components analysis, a strong effect of developmental stage (age) was seen (PC1). A likely effect of karyotype at each stage was generally seen in PC2, and more pronounced in the Brain-Seq 2 dataset (Fig. 1d). A noticeable divergence of "clustering" in the weeks following the onset of testosterone secretion (8wpc) in males was *not* obvious, suggesting most sex-dependent transcriptomic activity during this study period up until 15-17wpc is relatively fixed and predominantly reflects sex chromosome gene expression rather than major differences in sex hormone action at a global level. This hypothesis was supported by our global analysis (with n = 16 for each karyotype in both Brain-Seq 1 and Brain-Seq 2 datasets), as well as stage specific analyses, which found that most consistently differentially expressed genes (46,XX > 46,XY and 46,XY > 46,XX) across the time course were encoded by the sex chromosomes: that is, X chromosome genes involved in X inactivation (*XIST*, *TSIX*) in 46,XX samples (as well as *JPX* and *ZPX* in the global dataset), and a core

**Fig. 3 | Volcano plots for differentially expressed genes between 46,XX and 46,XY samples at each developmental stage. a** Brain-Seq 1 dataset. **b** Brain-Seq 2 dataset. At each developmental stage there are $n = 4$ samples in the 46,XX group and $n = 4$ samples in the 46,XY group. Red indicates adjusted $p$-value < 0.001; purple indicates adjusted $p$-value < 0.01; blue indicates adjusted $p$-value < 0.05. The ten genes with the highest -log10 adjusted $p$-value in each group are labeled (where adjusted $p$-value < 0.001). CS Carnegie stage, wpc weeks post conception.

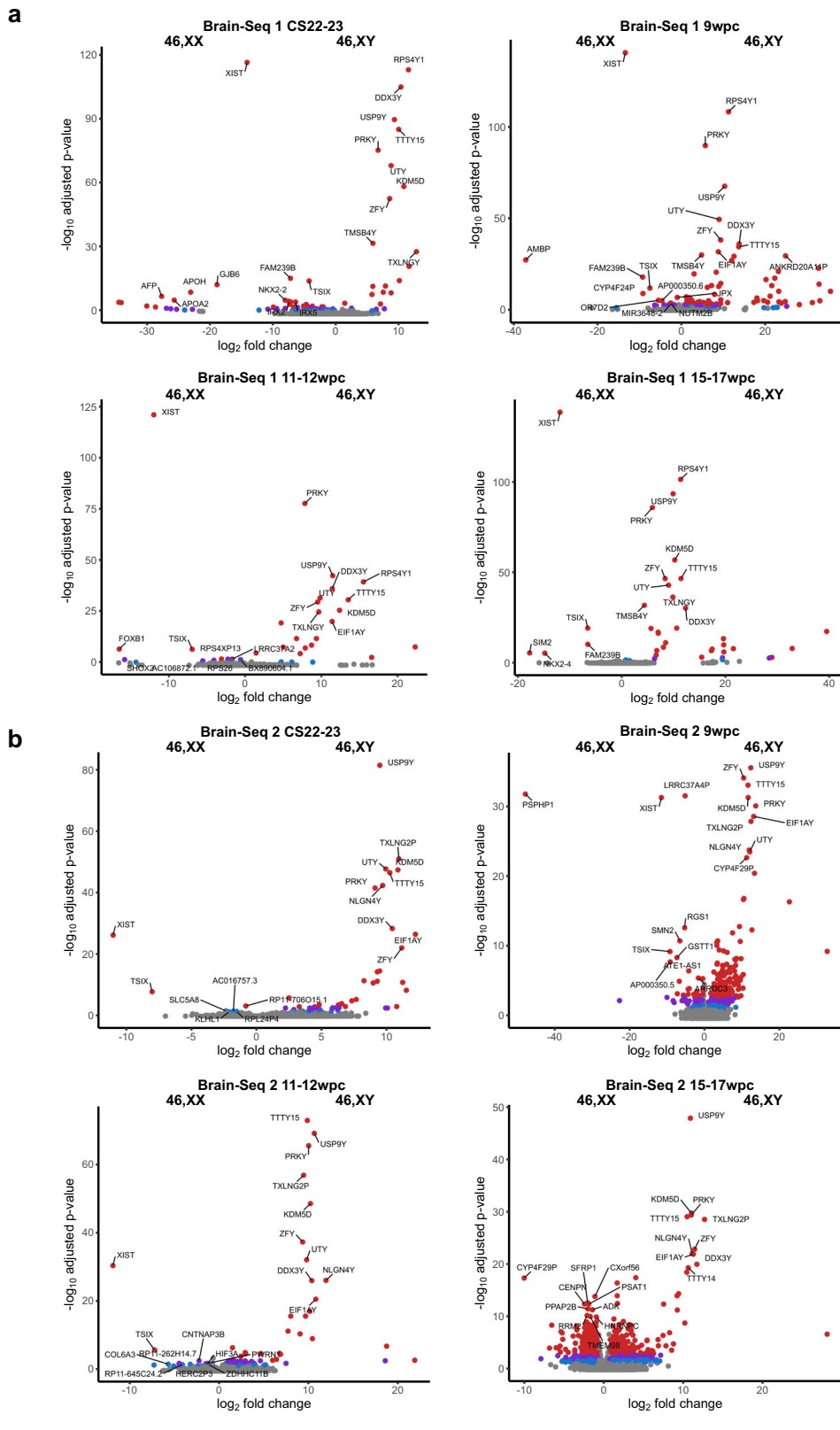

group of Y chromosome genes in 46,XY samples (Fig. 2 and Fig. 4; Supplementary data1_Brain-Seq 1; Supplementary data2_Brain-Seq 2). No autosomal gene differences were consistently seen, using the nominal adjusted $p$-value of <0.05 and during the developmental stages studied, although we cannot exclude that these could be seen with time or if even larger sample numbers had been included. Furthermore, when a supplementary analysis of sex differences in development was performed in other tissues (kidney, pancreas, liver, skin), somewhat similar patterns of differences in the principal components for 46,XX and 46,XY was seen (Supplementary Fig. 1). The differences in each of these tissues also mostly represented core X chromosome and Y chromosome genes. Taken together, global sex differences in early brain (cortex) gene expression were seen, but

**Fig. 4 | Sex-specific differential gene expression patterns during different stages of human brain development. a–d** Venn diagrams showing the overlap of differentially expressed genes (DEGs) in both RNA-seq datasets at each developmental stage (CS22-23; 9wpc; 11-12wpc; 15–17wpc). **a** 46,XX Brain-Seq 1 dataset; (**b**) 46,XY Brain-Seq 1 dataset; (**c**) 46,XX Brain-Seq 2 dataset; (**d**) 46,XY Brain-Seq 2 dataset. Differential expression was defined as an adjusted *p*-value < 0.05. Those genes found to be shared across all four stages in each dataset were identified and then compared between the two datasets. **e** Common 46,XX differentially expressed genes in both datasets. **f** Common 46,XY differentially expressed genes in both datasets. CS Carnegie stage, wpc weeks post conception.

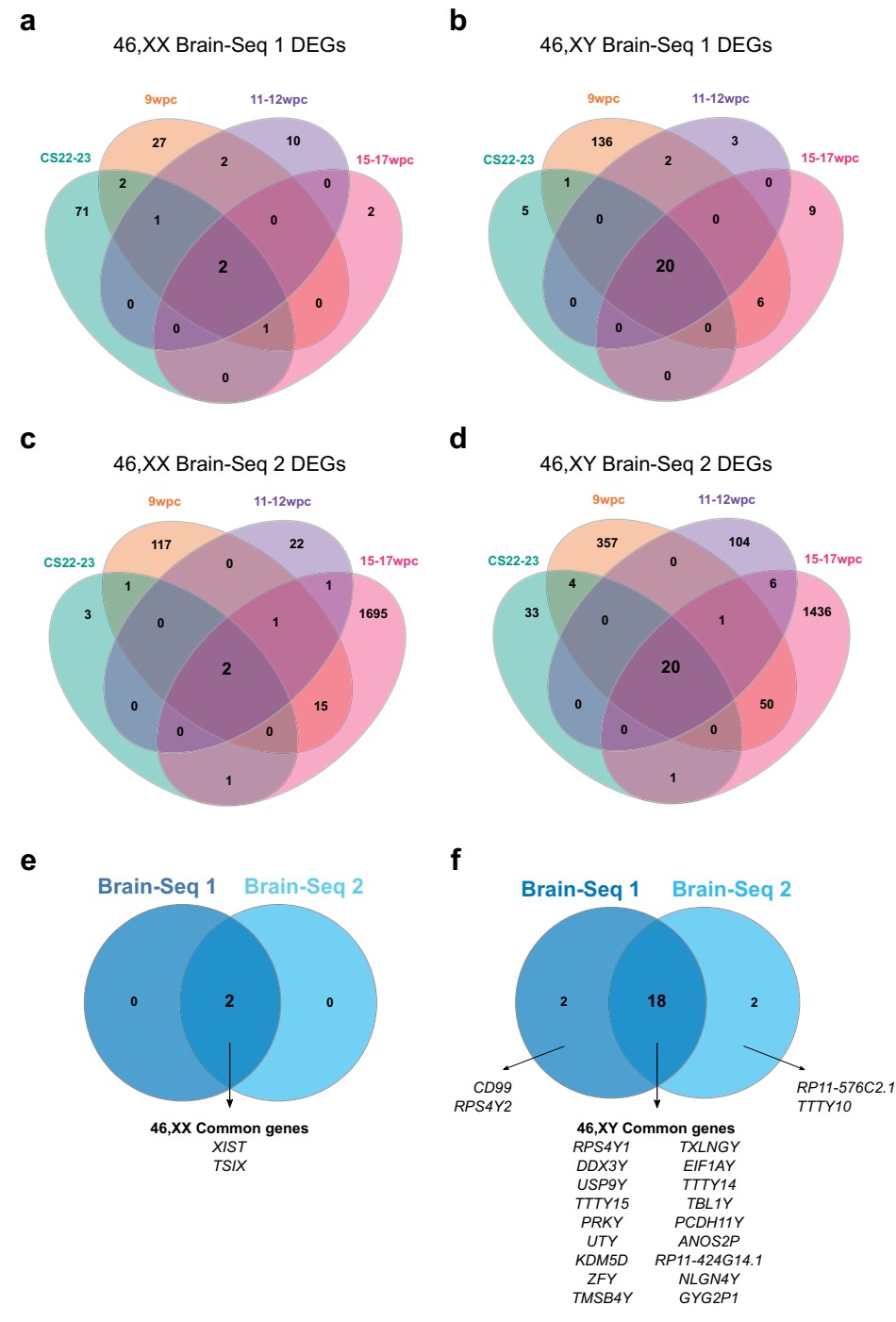

these largely reflected sex chromosome gene effects during this developmental time period. These findings reinforce previous microarray studies in developing human fetal brain, in which the largest sex differences were due to Y chromosome genes in the 46,XY brain, and where strong differential expression of *XIST* in 46,XX tissue was seen[23,24].

In order to identify potential brain-specific genes, we intersected our differentially expressed brain genes with those differentially expressed in other tissues. The two X chromosome genes involved in X inactivation (*XIST*, *TSIX*) were clearly differentially expressed in all 46,XX tissues, as expected, and were not brain specific. Of the 18 Y chromosome genes identified, only two emerged as highly brain specific; *PCDH11Y* and the non-coding *RP11-424G14.1* (ENSG0000260197) (Fig. 5a). Of note, several other Y chromosome genes showed strong brain expression in both datasets, but also had variable expression in other tissues (brain-enriched). Of

course, these genes could also have important developmental effects in the brain and in different organs, but also did not in general show strong brain specificity in adult transcriptomic datasets. Therefore, for the purposes of this study, we chose to focus mainly on the likely brain-specific genes.

The most relevant brain-specific Y chromosome gene identified was *PCDH11Y*. *PCDH11Y* encodes "protocadherin 11 Y-linked" (OMIM 400022), which was previously highlighted as a potentially important gene in 46,XY fetal brain development by Kang et al.[24] and Weickert et al.[22]. *PCDH11Y* encodes a protein that is proposed to act as a transmembrane adhesion molecule thought to be involved in neural development and synaptic cell-cell communication. *PCDH11Y* is expressed in adult brain datasets, and single cell analysis suggests that the transcript clusters in neuronal excitatory/inhibitory synaptic neurons (Human Protein Atlas). *PCDH11Y* is located on the short arm of the Y chromosome (p11.2) in a

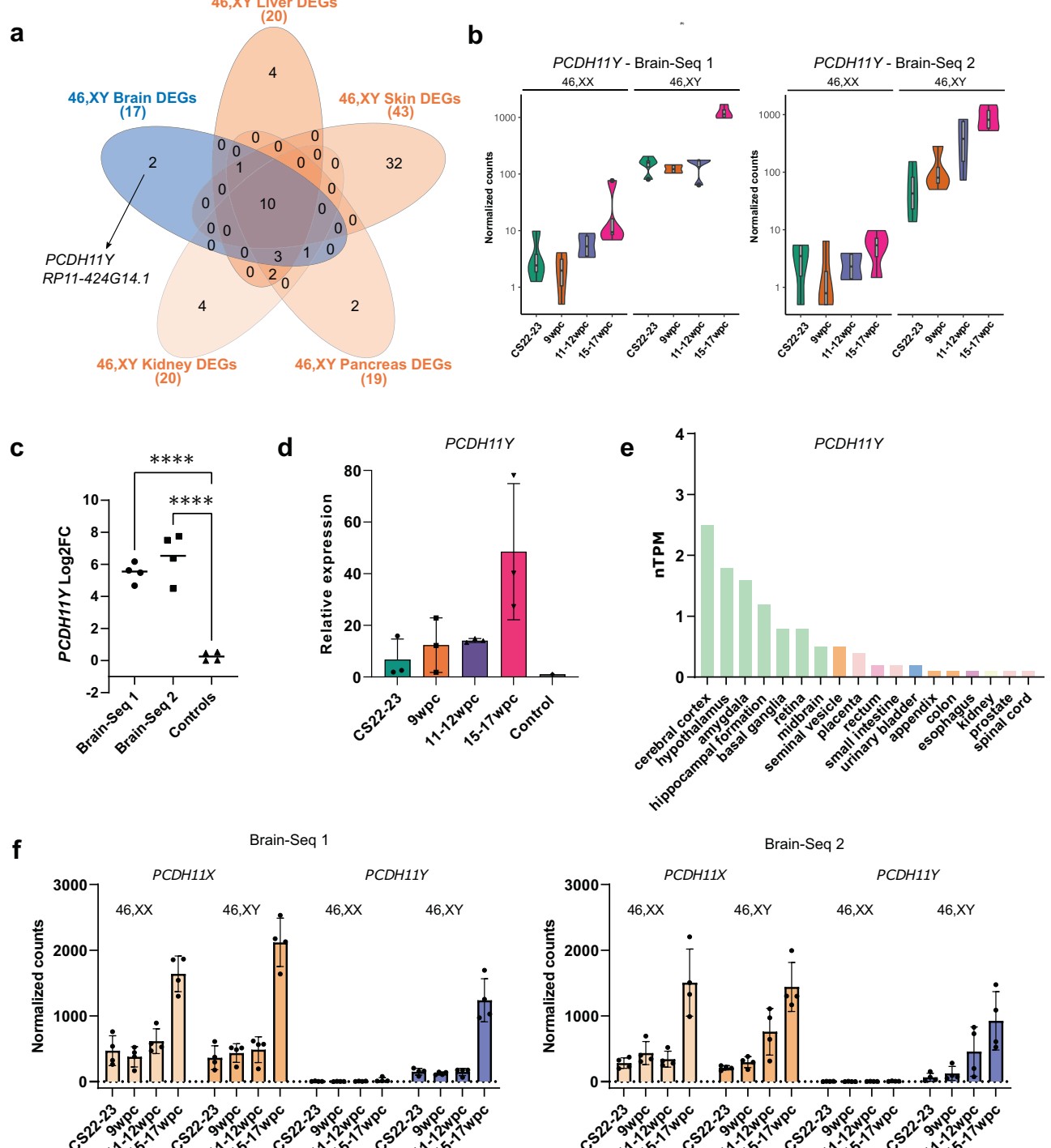

**Fig. 5 | Analyses of brain-specific genes in 46,XY cortex. a** Venn diagram showing the overlap of 46,XY differentially expressed genes (DEGs) common to both brain datasets and several 46,XY independent control tissues. **b** Violin plots of normalized counts showing expression patterns of *PCDH11Y* across developmental stages in the 46,XX and 46,XY brain and in both datasets (*n* = 4 in each stage). **c** Log2 fold change (Log2FC) values of *PCDH11Y* in both brain datasets compared to controls across each developmental stage (*n* = 4 in each group); horizontal line represents the median value. Statistical analysis was performed using a one-way ANOVA with Dunnett's multiple comparison tests; Brain-Seq 1 versus Controls, ****$p$-value < 0.0001; Brain-Seq 2 versus Controls, ****$p$-value < 0.0001. **d** qRT-PCR analysis of *PCDH11Y* in 46,XY brain cortex across key developmental stages, and compared to control. Two separate experiments were performed; in each, three independent samples were used for each stage, each done in triplicate. Four control tissues were used (kidney, liver, pancreas, skin). Representative data from one experiment is shown as mean with standard deviation. **e** Expression of *PCDH11Y* in the Human Protein Atlas Consensus dataset for adult brain, showing highest expression in cerebral cortex and hypothalamus, as well as in other brain structures (data accessed and downloaded from https://www.proteinatlas.org/ENSG00000099715-PCDH11Y/tissue). nTPM, normalized transcript per million. **f** Relative expression of *PCDH11X* and *PCDH11Y* in the Brain-Seq 1 and Brain-Seq 2 datasets (*n* = 4 in each group); data are shown as mean with standard deviation. CS Carnegie stage, wpc weeks post conception.

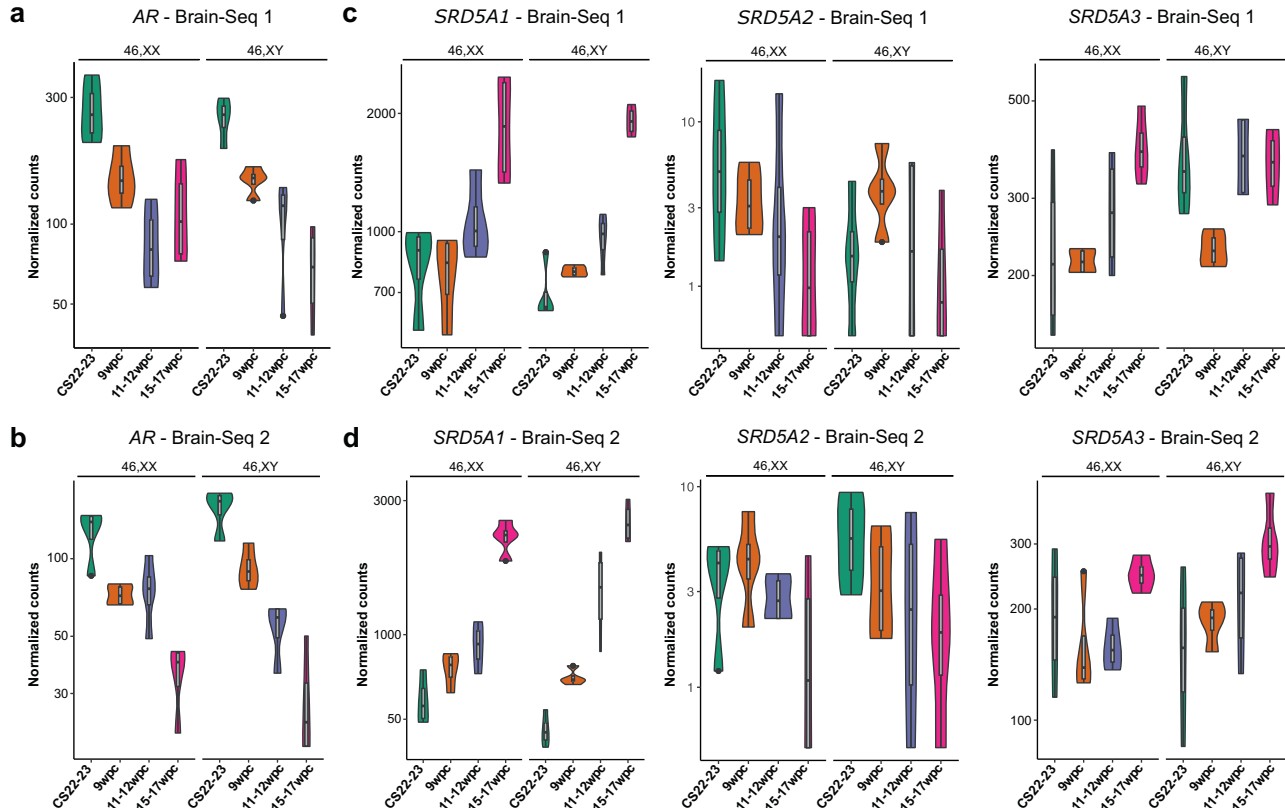

**Fig. 6 | Expression of genes encoding the androgen receptor (*AR*) and related factors involved in dihydrotestosterone biosynthesis. a** Violin plots of normalized counts showing expression patterns of the *AR* in the Brain-Seq 1 dataset (*n* = 4 in each group) and (**b**) Brain-Seq 2 dataset. **c** Violin plots of normalized counts showing expression patterns of the *SRD5A1*, *SRD5A2* and *SRD5A3* enzyme-encoding genes in the Brain-Seq 1 dataset (*n* = 4 in each group) and (**d**) Brain-Seq 2 dataset. CS Carnegie stage, wpc weeks post conception.

region with strong X chromosome homology ("X-degenerate region"). Indeed, *PCDH11Y* has 98% nucleotide similarity to its paralogue *PCDH11X* (Supplementary Fig. 3).

To date it has been very challenging to study differential protein expression between PCDH11Y and PCDH11X using immunohistochemistry because of the marked structural similarities between the proteins and limited differences in antigenicity. Studies using antibodies that do not differentiate between paralogues have been reported[37]. Others have studied gene level expression of *PCDH11Y* and *PCDH11X* in early human brain using a padlock probing and rolling circle amplification strategy [38]. Here, expression of *PCDH11Y* was found in spinal cord and medulla oblongata sections of only 46,XY human embryos (7-11 gestational weeks), and specifically in differentiating motor neuron cells. Of note, as in our data, similar levels of *PCDH11X* expression were seen in both 46,XY and 46,XX samples, whereas the expression of *PCDH11Y* appeared additive in 46,XY brain. These findings are important because they suggest that combined gene dosages of the two paralogues are not equal, and that *PCDH11Y* likely has a unique sex specific effect in the developing 46,XY brain.

The potential biological role of *PCDH11Y* is also relevant to sex differences. *PCDH11Y* (together with *PCDH11X*) has been implicated in cerebral hemisphere asymmetry in humans[39]. *PCDH11Y* has been associated with language development in a boy with a Y chromosome deletion involving this gene/locus[40]. A further study reported a child with severe language impairment and autistic behavior with a partial deletion of Y chromosome and consequent loss of *PCDH11Y* and *NLGN4Y*[41], which is another "brain enriched" Y chromosome gene arising from our analyses but with a less brain-specific expression pattern, previously reported to show specific expressing patterns in the postnatal human brain[28]. An additional link between *PCDH11Y* and autistic spectrum disorder (ASD) has been proposed in a small exome-wide population study[42].

Interestingly, *PCDH11Y* is unique to the human genome having evolved from the X homologue approximately 6 million years ago and it is not present in higher primates, such as chimpanzee or gorilla[43] (Supplementary Fig. 3). This locus, known as the "X-degenerate region", has a relatively recent evolutionary history and is distinct from the pseudoautosomal regions. Rather, it moved from an autosome to the sex chromosomes after the split between New and Old World monkeys. There remains high conservation of gene content and synteny across multiple genes between the X and Y chromosomes here, even though they have not recombined for several million years. There has also been significant gene inactivation and loss in different Old World monkey and ape genomes. This evolutionary history, together with the likely role of PCDH11Y in neuronal cell communication, makes *PCDH11Y*/PCDH11Y a potentially important candidate factor to study further in relation to sex differences in human language development as well as ASD.

The other brain-specific Y chromosome gene identified was *RP11-424G14.1*. This gene encodes a long non-coding RNA (lncRNA) and is located on the long arm of the Y chromosome. Long non-coding RNA transcripts sometimes have a regulatory role on genes in the region. The gene with closest proximity to *RP11-424G14.1* is *KDM5D*, but both have a minus strand orientation and *RP11-424G14.1* is 3' to *KDM5D*. Very little is known about any potential role of *RP11-424G14.1*, although long non-coding RNAs in general are emerging as potential regulators of transcriptomics in the human brain, and are one mechanism that might contribute to higher function in humans compared to other species[44–47]. Furthermore, it is likely that several of the core group of "brain-enriched" Y chromosome genes also play important roles in fetal brain development, as well as in other tissues where they show strong or differential expression (Supplementary Table 1; Supplementary Fig. 6). Tissue specific genes often play a key role in developmental fate, but larger networks of tissue-enriched/

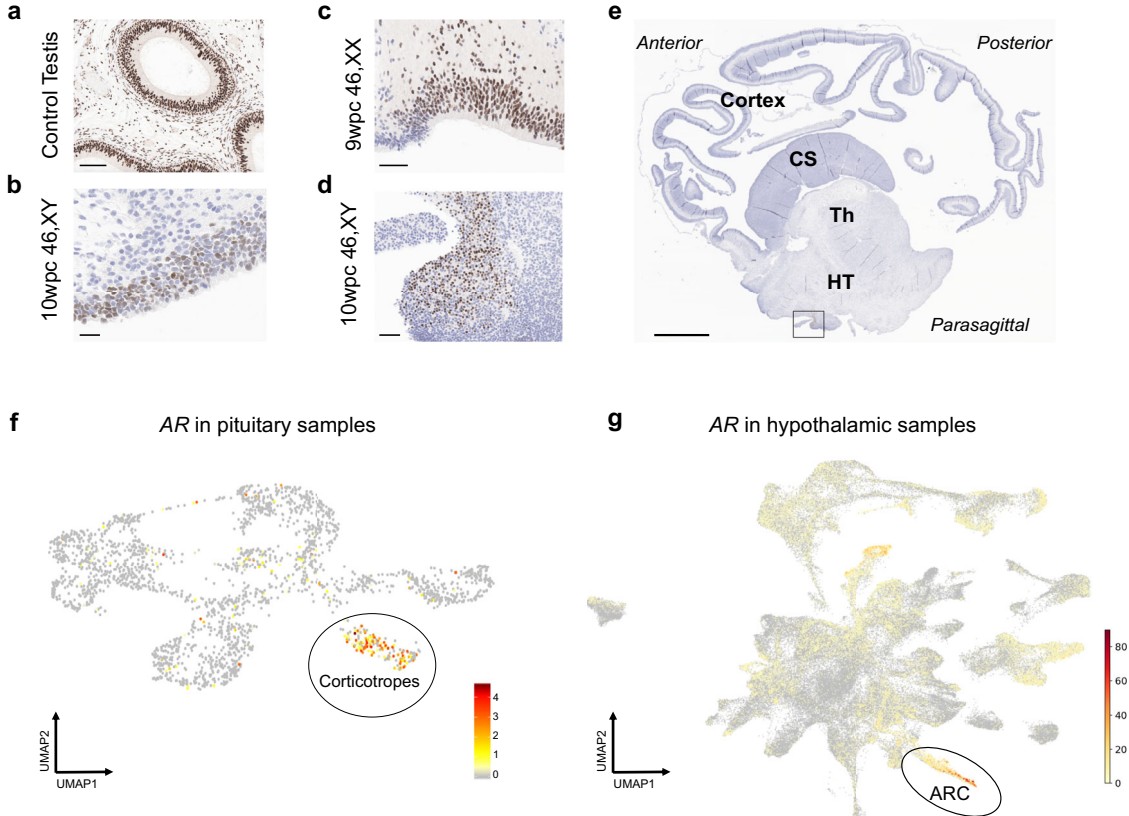

**Fig. 7 | Androgen receptor expression in the developing human brain.**
**a** Immunohistochemistry (IHC) of nuclear androgen receptor expression in control prepubertal testis (epididymis) (scale bar 100 μm) and **b** in the 10 week post conception (wpc) cerebral cortex (46,XY) (scale bar 20 μm). **c** IHC of localized regions of androgen receptor expression in the 9wpc cortex (46,XX) and **d** in a hypothalamic region at 10wpc (46,XY) (scale bars 50 μm), indicated by the rectangle in **e** which shows the orientation of this localized androgen receptor staining in the 10wpc brain is indicated by the rectangle in a parasagittal section scale bar 2 mm. CS, corpus striatum/ganglionic eminence; HT, hypothalamus; Th, thalamus. **f** Single cell RNA-seq expression of androgen receptor (*AR*) in human pituitary samples (7–25wpc)

(total *n* = 21; 46,XX *n* = 11; 46,XY *n* = 10). High expression of androgen receptor is shown in red in pituitary corticotropes (Supplementary Fig. 17). Data accessed from https://tanglab.shinyapps.io/Human_Fetal_Pituitary_Endocrine_Cells/ (CC BY 4.0, https://creativecommons.org/licenses/by/4.0/)[35]. **g** Single cell RNA-seq expression of androgen receptor (*AR*) in human hypothalamus samples (4–23wpc) (total *n* = 11; 46,XX *n* = 4; 46,XY *n* = 7). High expression of androgen receptor is shown in red in the arcuate nucleus (ARC) of the hypothalamus (Supplementary Fig. 18). Data accessed from https://nemoanalytics.org/p?l=a856c14e (CC BY-NC 4.0, https://creativecommons.org/licenses/by-nc/4.0/)[36]. UMAP, uniform manifold approximation and projection.

**Table 2 | Overview of AR expression in published single cell RNA-Seq studies of human brain development**

| Publication | Link to resource | Developmental stages Brain area | Expression level | Specific cell population |
|---|---|---|---|---|
| Xu et al., 2022[58] | https://heoa.shinyapps.io/base/ | 4–6wpc | Very low | N/A |
| Trevino et al., 2021[59] | https://scbrainregulation.su.domains | 16–24wpc | Low | N/A |
| Herb et al., 2023[36] | https://nemoanalytics.org/p?l=a856c14e | 4–23wpc Hypothalamus | Low | High in ARC |
| Zhang et al., 2020[35] | https://tanglab.shinyapps.io/Human_Fetal_Pituitary_Endocrine_Cells/ | 7–25wpc Pituitary | Low | Medium/high in corticotropes |

Data were taken from publicly available resources. ARC arcuate nucleus. N/A not available.

expressed factors are needed to facilitate complex developmental processes overall.

In addition to possible effects of sex chromosome genes, the other major mechanism of potential sex differences is through the effects of sex-dependent gonadal hormones, such as androgens (e.g. testosterone, dihydrotestosterone) or estrogens. As outlined above, the testis forms and starts to synthesize and release testosterone from around 8wpc in humans[13]. Testosterone has a direct effect on some developing tissues (e.g. Wolffian structures), whereas it is converted to the more potent hormone, dihydrotestosterone (DHT) by the enzyme 5α-reductase type 2 (encoded

by *SRD5A2*), in order to have an effect on other tissues (e.g. genital tubercle). All estrogens are synthesized from testosterone by the enzyme aromatase (*CYP19A1*). The developing human ovary is believed to be relatively quiescent during early development, with limited—if any—estrogen release.

The role of sex steroids in early human brain development is still unclear. Sex steroids likely mediate several biological differences in brain development, especially in relation to sexually dimorphic regulation of endocrine systems. The main sex hormone receptors (androgen receptor, AR; estrogen receptors, ESR1 (ERα), ESR2 (ERβ)) are more highly expressed

in the hypothalamus and pituitary in adults, as well as the hippocampus[48], and likely mediate their endocrine effects through these regions.

Despite the lack of direct studies on human fetal brain development, in vitro data have recently highlighted the central role that androgen action could have on early mechanisms of brain development. Indeed, exposing human cerebral organoids to androgens (testosterone and DHT), caused an increase in cortical progenitor proliferation (that develop into excitatory cortical neurons) and an increase in the neurogenic pool[49]. Treating mouse embryonic neural stem cells with testosterone lead to sex-specific changes in gene expression; in particular, an enrichment in epigenetic regulators was observed[50]. This work demonstrates how early hormonal influence potentially impacts brain development and might have long-term effects on sex-specific conditions.

Currently, very few data are available for AR expression during early human brain development, either at a transcript or protein level. In our study, we found relatively high and comparable expression of the androgen receptor gene (*AR*) in the brain cortex in both 46,XY and 46,XX fetuses at the time when the testis starts to release testosterone. These findings were supported by IHC, which showed regions of nuclear AR expression in the developing brain cortex. At a transcriptomic level, we observed a subsequent gradual decline in *AR* RNA expression in both sexes, and in both Brain-Seq 1 and Brain-Seq 2 datasets, up until the end of the observation period at 15-17wpc. Results were remarkably consistent between both 46,XY and 46,XX samples, and in both datasets, and were not expected as androgens often lead to increased expression of the androgen receptor. These findings support a general loss of global brain cortex *AR* expression in both the 46,XX and 46,XY brain at a time following the onset of testosterone secretion in the 46,XY fetus, leading to the low transcript levels as seen in adults. The teleological role for this pattern, if any, is currently unclear.

In order to identify potential androgen-responsive genes that could account for sex differences in the brain cortex, we sought genes that were consistently differentially expressed in 46,XY brain following testosterone (early-response, 9wpc vs CS22/23; sustained response, 15-17wpc vs CS22/23; p-value < 0.05) and that did not increase with time in a similar 46,XX analysis (thereby removing general cortex development genes), or where there was differential expression in the 46,XY compared to 46,XX tissue. We only considered genes in both the Brain-Seq 1 and Brain-Seq 2 datasets, in order to have higher stringency. Although several candidate genes emerged, none showed time-series expression patterns consistent with changes in androgens in the 46,XY fetal brain cortex.

In addition to more generalized brain cortex effects, it is established that androgens and the androgen receptor can have more specific or defined roles, for example in endocrine systems. Using IHC of human fetal brain, we were able to show localized regions of AR expression, such as in the inferior hypothalamus. We linked this to emerging data from single cell RNA-sequencing, which identified *AR* expression in more localized structures such as the corticotropes of the pituitary gland[35] (involved in adrenal and stress responses) and arcuate nucleus of the hypothalamus[36] (implicated in appetite regulation and reproduction). Other datasets, such as the Allen Brain database (https://www.brainspan.org/rnaseq/search/index.html), show some potential *AR* expression in the hippocampus, but transcript levels are low, and *AR* expression in most brain regions is low in the adult, without major sex differences (Supplementary Fig. 7). Taken together, a general reduction in global *AR* expression in the brain cortex occurs in both sexes during the late first and early second trimesters, but subsequent localization of AR expression (and action) to key regions is likely to be important and requires further investigation.

Relatively little is known about early human fetal expression of pathways involved in conversion of testosterone to DHT (by 5α-reductase enzymes, *SRD5A*) or to estrogens (by aromatase, *CYP19A1*), or in neurosteroidogenesis (Supplementary Figs. 10 and 11). Indeed, questions remain about the fetal blood brain barrier and the dynamics of secreted gonadal sex steroids reaching target tissues of interest. Again, using a global approach of brain cortex bulk RNA-Seq we were unable to detect significant expression

of the key enzyme responsible for conversion of testosterone to DHT, 5α-reductase type 2 (*SRD5A2*) (Fig. 6c, d), nor significant expression of genes involved in estrogen synthesis and action, *CYP19A1*, *ESR1* and *ESR2* (Supplementary Fig. 12). Relatively high expression of the enzymes 5α-reductase type 1 (*SRD5A1*) and 5α-reductase type 3 (*SRD5A3*) were observed in both sexes (Fig. 6c and d), although these have much lower affinity for the conversion of testosterone to DHT. Although several genes involved in classic and "backdoor" pathways of androgen biosynthesis were expressed, the potential "gatekeeper" in both these systems – *CYP17A1* – was not expressed (Supplementary Figs. 10 and 11). As with the androgen receptor, more localized effects in key regions or nuclei of the brain may be important.

This study has several limitations. Firstly, the anatomy of the developing cortex is complex and changes over time, and standardized sampling of fetal material is challenging. By using a bulk RNA-seq approach we have likely reduced sampling bias compared to single cell approaches, at the expense of detail, but we did generate remarkably consistent results from two independent datasets. Second, even with four samples in each karyotype group, we may have been underpowered to detect more subtle changes in the analysis across time. Having two replication datasets was useful to maintain a stringent approach, but transcriptomic sex differences are often associated with lower fold change differences, so large sample numbers are needed. Despite this, even with our global study (n = 16 in each group), the main consistent differences were in sex chromosome genes. We did not attempt to address sex-specific isoforms[51]. Third, immunohistochemical analysis of *PCDH11Y* is very difficult given its similarity to *PCDH11X*. Finally, we focused mostly on global brain cortex differences over time, but more focused analysis of key regions and nuclei will provide more localized information in the future, once more specific markers are known.

Despite these limitations, this is one of the first studies to specifically address sex differences in early human fetal brain development between the late first and early second trimesters, and at a time of potentially important changes in sex hormone exposure. This work should help to focus future efforts in the field as new technologies and strategies emerge.

## Methods
### Inclusion and ethics
Human embryonic and fetal cerebral cortex samples used for bulk RNA-sequencing (RNA-seq) were obtained with ethical approval and with informed consent from the Medical Research Council (MRC)/Wellcome Trust-funded Human Developmental Biology Resource (HDBR) (http://www.hdbr.org) (Research Ethics Committee references: 08/H0712/34 + 5, 18/LO/0822, 08/H0906/21 + 5, 18/NE/0290; HDBR project references 200332, 200655). The HDBR is a biobank that is regulated by the UK Human Tissue Authority (HTA; www.hta.gov.uk) and acts within the HTA governance framework. All ethical regulations relevant to human research participants were followed.

### Samples
Fetal tissues were collected following medical termination of pregnancy (Mifepristone), except for three early samples collected following ultrasound-guided aspiration and seven (Brain-Seq 1) obtained following standard surgical procedures. After rinsing, samples were placed in L15 (Liebovitz) medium (Sigma-Aldrich) on ice for transport. Mean times in transit were: Brain-Seq 1, 46,XY, 168 min; Brain-Seq 1, 46,XX, 116 min.

Samples were dissected further for regions of interest following the protocols outlined in detail for the Brain-Seq 2 study (Linday et al., Front Neuroanat[26]). In brief, the forebrain was initially dissected from the mid-brain and hindbrain. In earlier stages (CS22-23), the telencephalon (Fig. 1a) was separated from the diencephalon, and sampled. In later stages (9-17 wpc), the cerebral cortex (Fig. 1a) was separated from the diencephalon and coronal strips taken. Samples were stored at -70°C until RNA extraction. For immunohistochemistry, cortex samples or whole brain hemisphere samples were fixed in 10% formalin prior to processing.

The age of embryos up to 8wpc was calculated based on Carnegie staging (CS), whereas in fetuses (>8wpc) the age was estimated from knee-heel length and foot length in relation to standard growth data. Karyotyping was performed by G-banding or quantitative PCR (chromosomes 13, 16, 18, 21, 22 and X and Y) to determine the sex of the embryo/fetus as well as to exclude any major chromosomal rearrangements or aneuploidies.

Data comparing differences between 46,XX and 46,XY tissues (liver, kidney, skin, pancreas) (bulk RNA-Seq) were generated, using the same bioinformatic pipeline described below, as part of a project into the developmental effects of sex chromosomes (HDBR project reference 200581) (accession number E-MTAB-13673; https://www.ebi.ac.uk/biostudies/ArrayExpress/studies/E-MTAB-13673?query=E-MTAB-13673)[52].

## Bulk RNA-sequencing
Bulk RNA-Seq was undertaken in two independent datasets that are described in detail below. An overview of all samples included in these studies is outlined in Table 1.

**Brain-Seq 1 dataset.** In this study, total RNA was extracted from newly acquired human embryonic/fetal brain samples (*n* = 32) using the All-Prep DNA/RNA Mini Kit (Qiagen), according to the manufacturer's instructions. RNA quality and concentration were assessed using a Tapestation 4200 platform (Agilent, California, USA). RNA Integrity Numbers (RIN) ranged from 7.2 to 9.5. Importantly, there were no significant differences between RINs for 46,XY samples (mean 8.5, range 7.5–9.5) and 46,XX samples (mean 8.3, range 7.2–9.3). There were no significant differences between RINs at each stage (CS22, mean 8.5; 9wpc, mean 8.2; 11wpc, mean 8.6; 15wpc, mean 8.5) (Supplementary data 1.1_Brain-Seq 1). cDNA libraries were prepared using the KAPA mRNA HyperPrep Kit (Roche) and subsequently sequenced on a NextSeq 500 sequencer (paired-end 43 base pairs) (Illumina, San Diego, CA) in a single run to reduce potential batch effects. Fastq files were processed by FastQC (Babraham Bioinformatics) and aligned to the Human Genome (Ensembl, GRCh37) using STAR (2.5.2a)[53]. The matrix containing uniquely mapped read counts was generated using featureCounts[54], part of the R package Rsubread. MultiQC was used to visualize results (Supplementary Fig. 19). LFC shrinkage function was applied before performing differential-expression analysis using DESeq2, following standard methods[55].

**Brain-Seq 2 dataset.** Bulk RNA-seq data for an independent parallel replication study (Brain-Seq 2) were obtained from a Human Developmental Biology Resource (HDBR) project focusing on early human brain development (https://www.ebi.ac.uk/gxa/experiments/E-MTAB-4840/Downloads)[26]. Samples were prepared and extracted using the approaches described above, and consistent with Brain-Seq 1. Library preparation and sequencing (IIlumina HiSeq2000) are described previously (Lindsay et al., Front Neuroanat[26]). Raw fastq files were obtained and analyzed using the same bioinformatic pipeline as Brain-Seq 1, as described above. To achieve a balanced experimental design and match the developmental stages to those used in the Brain-Seq 1 dataset, *n* = 8 samples per stage (four 46,XX, four 46,XY) were included from the Brain-Seq 2 dataset (Table 1; Supplementary data 2.1_Brain-Seq 2).

## Quantitative reverse transcriptase PCR (qRT-PCR)
Purified RNA was quantified using a NanoDrop 1000 spectrophotometer (Thermo Fisher Scientific). RNA (2 µg) was reverse transcribed with a SuperScript III Reverse Transcription kit (Applied Biosystems, Thermo Fisher Scientific) according to the manufacturer's instructions. qRT-PCR was performed using TaqMan Fast Advanced Master Mix (Applied Biosystems) and TaqMan gene assays (*AR* Hs00171172_m1; *PCDH11Y* Hs00263145_m1, Thermo Fisher Scientific) on a StepOnePlus System (Thermo Fisher Scientific). The relative expression of gene was calculated as $2-\Delta\Delta Ct$ using the comparative Ct ($\Delta\Delta Ct$) method and GAPDH (Hs02758991_g1) (Thermo Fisher Scientific) as an internal housekeeping

control. Experiments were repeated on two independent occasions; three samples per developmental stage were used, each performed with triplicate replicates. Four control tissues were used (kidney, liver, pancreas, skin). Representative data of one experiment are shown.

## Adult gene expression data
Expression data for key genes of interest in adult tissues was obtained from the Human Protein Atlas (HPA) Consensus expression data (v23) (https://www.proteinatlas.org/)[5] and from the Genotype-Tissue Expression (GTEx) Project (https://gtexportal.org/home/). The data used for the analyses described in this manuscript were obtained from the HPA and GTEx Portal on 10/10/23.

## Genomic characterization
The genomic location and structure of genes of interest was obtained from Ensembl Genome Browser 110 accessed on 10/10/23 (https://www.ensembl.org/index.html)[12] (GRCh38.p14)[12]. Species comparisons for PCDH11Y were generated using GeneTree (ENSGT00940000158335) in the Ensembl Genome Browser.

## Immunohistochemistry (IHC)
Human brain cortex samples (46,XY) at 10wpc and 17wpc were fixed in 10% formalin for 24–48 h before being processed, embedded and sectioned for histology and immunohistochemistry (IHC). Hematoxylin and eosin (H&E) staining was performed using standard methods on 3 µm sections. IHC was performed on a Leica Bond-max automated platform (Leica Biosystems). In brief, antigen retrieval was undertaken to unmask the epitope (Heat Induced Epitope Retrieval (HIER), Bond-max protocol F), then endogenous activity was blocked with peroxidase using a Bond polymer refine kit (cat # DS9800). Next, slides were incubated with a primary androgen receptor (AR, NR3C4) antibody for 1 h (Abcam ab108341 ChIP grade, 1:50 dilution, HIER2 for 20 min) before having a post-primary antibody applied and horseradish peroxidase (HRP) labelled polymer, followed by 3, 3-diaminobenzidine (DAB) chromogen solution (all Bond polymer refine kit). Sections were counterstained with hematoxylin, washed, dehydrated in graded alcohols, cleared in two xylene changes and mounted. Imaging was undertaken using an Aperio CS2 Scanner (Leica Biosystems) at 40x objective. Subsequent analysis was performed with QuPath (v.0.2.3) (https://qupath.github.io) and Leica ImageScope (Leica Biosystems) software.

## Single cell RNA-sequencing (scRNA-Seq) data
scRNA-Seq data were obtained for androgen receptor (*AR*) expression using the following resources:

1) human fetal pituitary cell images were generated from the open resource (CC BY 4.0 license) on the shiny webpage https://tanglab.shinyapps.io/Human_Fetal_Pituitary_Endocrine_Cells/ developed by Zhang et al.[35].

2) human hypothalamus images were generated from the NeMo Analytic open resource (CC-BY-NC 4.0 International license) webpage https://nemoanalytics.org/p?l=a856c14e&g=gad2 developed by Herb et al.[36].

Data are represented as UMAP (uniform manifold approximation and projection, for dimension reduction) images for cell clusters at different stages of development.

## Graphical representation
Venn diagrams were produced using the online tool InteractiVenn (http://www.interactivenn.net/#)[56]. Graphics were generated using GraphPad Prism version 8.4.3 for Windows (GraphPad Software, San Diego, USA; www.graphpad.com).

## Statistical analysis and reproducibility
Statistical analyses were performed using GraphPad Prism version 8.4.3 for Windows (GraphPad Software, San Diego, USA; www.graphpad.com) and

R (RStudio 2022.07.2 for macOS), and are described in the relevant figure legends. A *p*-value < 0.05 was considered significant. In general, *n* = 4 samples were used for each stage/karyotype group and *n* = 16 for the "global" gene analysis. All studies and analyses were balanced for 46,XX and 46,XY samples, with stages/ages shown in Table 2. The generation of two independent datasets (Brain-Seq 1 and Brain-Seq 2) allowed for much greater reproducibility of data throughout this analysis. Additional reproducibility was provided in some situations by qRT-PCR and IHC.

## Reporting summary

Further information on research design is available in the Nature Portfolio Reporting Summary linked to this article.

## Data availability

The datasets generated and analyzed during the current study can be found in the following links:

1. Brain-Seq 1 (bulk RNA-seq): ArrayExpress/Biostudies (accession number E-MTAB-13662; https://www.ebi.ac.uk/biostudies/ArrayExpress/studies/E-MTAB-13662?query=E-MTAB-13662)

2. Brain-Seq 2 (bulk RNA-seq): https://www.ebi.ac.uk/gxa/experiments/E-MTAB-4840/Downloads [26].

3. Control samples (bulk RNA-seq): ArrayExpress/Biostudies (accession number E-MTAB-13673; https://www.ebi.ac.uk/biostudies/ArrayExpress/studies/E-MTAB-13673?query=E-MTAB-13673) [52].

4. Supplementary Data files (Brain-Seq 1, Brain-Seq 2; Control samples; Fig. 5 and Supplementary Fig. 8) have been deposited in Open Science Framework (https://doi.org/10.17605/OSF.IO/AJ3RS) [57]."

All other data are available from the corresponding author as applicable, on reasonable request."

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

## Acknowledgements

This research was funded in whole, or in part, by the Wellcome Trust (grant 209328/Z/17/Z). For the purpose of Open Access, the author has applied a CC BY public copyright license to any Author Accepted Manuscript version arising from this submission. We also thank other members of the Human Developmental Biology Resource and UCL Genomics for their additional contributions to this work. Human fetal material was provided by the Joint MRC/Wellcome Trust (Grant MR/R006237/1, MR/X008304/1 and 226202/Z/22/Z) Human Developmental Biology Resource (http://www.hdbr.org). All research at Great Ormond Street Hospital NHS Foundation Trust and UCL Great Ormond Street Institute of Child Health is made possible by the National Institute for Health Research Great Ormond Street Hospital Biomedical Research Centre. The views expressed are those of the authors and not necessarily those of the National Health Service, National Institute for Health Research, or Department of Health. The Genotype-Tissue Expression (GTEx) Project was supported by the Common Fund of the Office of the Director of the National Institutes of Health, and by NCI, NHGRI, NHLBI, NIDA, NIMH, and NINDS. The data used for the analyses described in this manuscript were obtained from the GTEx Portal on 10/10/23.

## Author contributions

Author contributions were as follows. Study conceptualization: F.B., I.d.V., J.C.A.; Methodology: F.B., J.P.S., I.d.V., J.C.A.; Investigation: F.B., J.P.S., O.K.O., A.J., N.M., P.N., T.B., N.S., M.T.D., I.d.V.; Formal analysis: F.B., J.P.S., I.d.V.; Data curation: F.B.; Resources: N.S.; Project administration: F.B., J.C.A.; Supervision: F.B., I.d.V., J.C.A.; Validation: F.B., I.d.V., J.C.A.; Visualization: F.B., O.K.O., J.C.A.; Writing—original draft: F.B., J.C.A.; Writing—review and editing: All authors; Funding acquisition: J.C.A.

## Competing interests

The authors declare no competing interests.
