## [Transparent Peer Review file · Communications Biology]

Transcriptomic sex differences in early human fetal brain development

Corresponding Author: Dr Federica Buonocore

This manuscript has been previously reviewed at another journal. This document only contains information relating to versions considered at Communications Biology.

Version 0:

Reviewer comments:

Reviewer #1

(Remarks to the Author)

Compliments to the authors for all the work they have done in response to the reviewers' feedback and the thorough revision of the manuscript, with addition of important information to all sections. I have no remaining comments or questions.

Reviewer #2

(Remarks to the Author)

The manuscript by Buonocore et al. analyses the bulk transcriptome of the human embryonic brain cortex at four stages spanning 7 to 17 weeks post-conception (pc) and reports key differences in gene expression between sexes (XX and XY genotypes).

Despite the higher quality of this dataset compared to previous studies of this type, primarily microarray-based, the study does not present groundbreaking findings. It confirms that XIST and TSIX lncRNAs are expressed in XX samples and that a set of Y-chromosome-specific genes is expressed in XY samples. The analysis of sex hormone effects also yields no notable new insights. The most valuable contribution is therefore likely the dataset itself.

The most interesting observations for me were:

- 1) No differential expression between XX and XY genotypes is induced by the onset of testosterone secretion at eight weeks pc, suggesting that all expression differences observed during this period are governed by transcriptional control linked to the sex chromosomes themselves.
- 2) There is no difference in autosomal gene expression, suggesting that none of the differentially expressed sex chromosome genes act as upstream regulators of autosomal gene expression in this region.

The authors have addressed most of the criticisms from the previous round of review (which I was not involved in). My only comments are as follows:

- ****Line 259****: The authors state, "Karyotype information for these two datasets was not available and therefore analyses of sex differences could not be carried out." However, the sex-specific transcripts identified in both this and previous studies should be more than sufficient to unambiguously determine the sex of each ssRNA-seq sample, even if this information was not explicitly included in the metadata. The worst-case scenario would be that all samples belong to the same sex.

- ****Minor (Line 163)****: The authors mention the "strong X chromosome homology" of the PCDH11Y gene and its locus. This should be explained more clearly. This locus, known as the X-degenerate region, has a fascinating and relatively recent evolutionary history. It is ****not**** part of the pseudoautosomal regions. Rather, it moved from an autosome to the sex chromosomes after the split between New and Old World monkeys. There remains high conservation of gene content and synteny across multiple genes between the X and Y chromosomes, even though they have not recombined for several million years. There has also been significant gene inactivation and loss in different Old World monkey and ape genomes.

Reviewer #3

(Remarks to the Author)

The authors have done an excellent job in responding to the extensive previous reviews. They now place their work appropriately in the context of the existing literature, have dropped the 1.5 fold-change filter and conducted a global analysis, among others, all of which improve the impact. They have also resisted the urge to draw conclusions using non-qualitative measures such as AR IHC and have avoided overly speculative conclusions.

**“Transcriptomic sex differences in early human fetal brain development”
*Buonocore et al.***

Reviewers' comments (in bold):

Reviewer #1 (Remarks to the Author):

Compliments to the authors for all the work they have done in response to the reviewers' feedback and the thorough revision of the manuscript, with addition of important information to all sections. I have no remaining comments or questions.

We thank the reviewer for their input.

Reviewer #2 (Remarks to the Author):

The manuscript by Buonocore et al. analyses the bulk transcriptome of the human embryonic brain cortex at four stages spanning 7 to 17 weeks post-conception (pc) and reports key differences in gene expression between sexes (XX and XY genotypes).

Despite the higher quality of this dataset compared to previous studies of this type, primarily microarray-based, the study does not present groundbreaking findings. It confirms that XIST and TSIX lncRNAs are expressed in XX samples and that a set of Y-chromosome-specific genes is expressed in XY samples. The analysis of sex hormone effects also yields no notable new insights. The most valuable contribution is therefore likely the dataset itself.

We appreciate this new reviewer's efforts, and agree that making these two replicated datasets available for the research community is important, as these data may provide rapid validation of future work in the field and for those working on human brain development in general.

The most interesting observations for me were:

- 1) No differential expression between XX and XY genotypes is induced by the onset of testosterone secretion at eight weeks pc, suggesting that all expression differences observed during this period are governed by transcriptional control linked to the sex chromosomes themselves.**
- 2) There is no difference in autosomal gene expression, suggesting that none of the differentially expressed sex chromosome genes act as upstream regulators of autosomal gene expression in this region.**

The authors have addressed most of the criticisms from the previous round of review (which I was not involved in). My only comments are as follows:

- ****Line 259****: The authors state, “Karyotype information for these two datasets was not available and therefore analyses of sex differences could not be carried out.” However, the sex-specific transcripts identified in both this and previous studies should be more than sufficient to unambiguously determine the sex of each ssRNA-seq sample, even if this information was not explicitly included in the metadata. The worst-case scenario would be that all samples belong to the same sex.

We thank the reviewer for this comment. With the available online tools provided to analyse these datasets we were not able to properly assign individual cells to karyotype/sex. However, we agree that looking for key differentially expressed genes is a useful surrogate marker for karyotype. We have gone back to available online tools and generated feature plots for UTY (to identify 46,XY cells) and XIST (to identify 46,XX cells). We have done this for both the pituitary and hypothalamus datasets. We have split Supplementary Figure 18 into two, and we now show these data together with the relevant cell annotation in Supplementary Figure 17 (for pituitary) and Supplementary Figure 18 (for hypothalamus). Using this approach we confirm that both datasets do include samples from 46,XY and 46,XX fetuses, consistent with the experimental design outlined, although there may be a stronger contribution of 46,XY cells overall.

UTY and XIST in the human pituitary:

UTY and XIST in the human hypothalamus:

AR expression appears to be potentially higher in a subset of 46,XY-derived cells. However, we feel that larger studies that are specifically designed and powered to detect sex differences are needed for this to be investigated in detail.

AR in the human pituitary:

AR in the human hypothalamus:

Based on these findings we have changed the text as follows:

ORIGINAL TEXT Results, line 259-262: “Taken together, these findings suggest that localized expression of the *AR* may be important with time. Karyotype information for these two datasets was not available and therefore analyses of sex differences could not be carried out. However, this is important and future studies should focus on more localized regions of high *AR* expression and potential sex differences in specific cell populations.”

REVISED TEXT Results, line 259-262: “These findings suggest that localized expression of the *AR* may be important with time. In addition, we generated feature plots in both datasets (pituitary, hypothalamus) for *UTY* (to identify 46,XY cells) and *XIST* (to identify 46,XX cells). This approach confirmed the presence of both 46,XY and 46,XX cells, and a potential higher expression of *AR* in 46,XY-derived cells (Supplementary Figures 18 and 19). Taken together, future studies should focus on more localized regions of high *AR* expression as well as on potential sex differences in specific cell populations.”

We have also removed the following sentence:

Discussion, lines 413-414: “although it is unclear whether sex differences occur as karyotype information on these data is not available.”

- **Minor (Line 163)**: The authors mention the “strong X chromosome homology” of the *PCDH11Y* gene and its locus. This should be explained more clearly. This locus, known as the X-degenerate region, has a fascinating and relatively recent

evolutionary history. It is ****not**** part of the pseudoautosomal regions. Rather, it moved from an autosome to the sex chromosomes after the split between New and Old World monkeys. There remains high conservation of gene content and synteny across multiple genes between the X and Y chromosomes, even though they have not recombined for several million years. There has also been significant gene inactivation and loss in different Old World monkey and ape genomes.

Thank you. We hope we did not imply that *PCDH11Y* is part of the pseudoautosomal region, but rather a gene with X-Y ancestral homology on the short arm of the Y chromosome. We agree that this region is fascinating in terms of evolution, which is why we provided detail about this in Supplementary Figure 3 and in the discussion. However, we have expanded this section to include the very helpful summary suggested by the reviewer, for which we are grateful.

We have made the following changes:

Results, original lines 162-165: "*PCDH11Y* is located on the short arm of the Y chromosome (p11.2) (Chr Y: 4,868,267-5,610,265, GRCh37; Chr Y:5,000,226-5,742,224, GRCh38) in a locus with strong X chromosome homology ("X-degenerate region"), that is separate from the pseudoautosomal regions. Although *PCDH11Y* is unique to humans, a related X chromosome gene (*PCDH11X*) exists (Supplementary Figure 3)."

Discussion, original lines 322-323: "*PCDH11Y* is located on the short arm of the Y chromosome (p11.2) in a region with strong X chromosome homology ("X-degenerate region")."

Discussion, original lines 343-345: "Interestingly, *PCDH11Y* is unique to the human genome having evolved from the X homologue approximately 6 million years ago and it is not present in higher primates, such as chimpanzee or gorilla⁴³ (Supplementary Figure 3). This locus, known as the "X-degenerate region", has a relatively recent evolutionary history and is distinct from the pseudoautosomal regions. Rather, it moved from an autosome to the sex chromosomes after the split between New and Old World monkeys. There remains high conservation of gene content and synteny across multiple genes between the X and Y chromosomes here, even though they have not recombined for several million years. There has also been significant gene inactivation and loss in different Old World monkey and ape genomes. This evolutionary history, together with the likely role of *PCDH11Y* in neuronal cell communication, makes *PCDH11Y/PCDH11Y* a potentially important candidate factor....."

Reviewer #3 (Remarks to the Author):

The authors have done an excellent job in responding to the extensive previous reviews. They now place their work appropriately in the context of the existing literature, have dropped the 1.5 fold-change filter and conducted a global analysis, among others, all of which improve the impact. They have also resisted the urge to draw conclusions using non-qualitative measures such as AR IHC and have avoided overly speculative conclusions.

Thank you. We are pleased the reviewer acknowledges the changes made. We really appreciate their initial comments, which we feel have strengthened the focus and rigor of our manuscript.